# The Aryl Hydrocarbon Receptor (AHR): Peacekeeper of the Skin

**DOI:** 10.3390/ijms26041618

**Published:** 2025-02-14

**Authors:** Hannah R. Dawe, Paola Di Meglio

**Affiliations:** 1St John’s Institute of Dermatology, King’s College London, London SE1 9RT, UK; hannah.dawe@kcl.ac.uk; 2KHP Centre for Translational Medicine, London SE1 9RT, UK

**Keywords:** aryl hydrocarbon receptor, skin, psoriasis, atopic dermatitis, tapinarof, skin inflammation

## Abstract

In the last decade, the aryl hydrocarbon receptor (AHR) has emerged as a critical peacekeeper for the maintenance of healthy skin. The evolutionary conservation of AHR implied physiological functions for this receptor, beyond the detoxification of man-made compounds, a notion further supported by the existence of physiological AHR ligands, notably derivates of tryptophan by the host and host microbiome. The UV light-derived ligand, 6-formylindolo[3,2-*b*]carbazole (FICZ), anticipated a role for AHR in skin, a UV light-exposed organ, where physiological AHR activation promotes a healthy skin barrier and constrains inflammation. The clinical development of tapinarof, the first topical AHR modulating drug for inflammatory skin disease, approved by the FDA for mild-to-moderate psoriasis and poised for approval in atopic dermatitis, supports the therapeutic targeting of the AHR pathway to harness its beneficial effect in skin inflammation. Here, we describe how a tightly controlled, physiological activation of the AHR pathway maintains skin homeostasis, and discuss how the pathway is dysregulated in psoriasis and atopic dermatitis, identifying areas offering opportunities for alternative therapeutic approaches, for further investigation.

## 1. Introduction

The aryl hydrocarbon receptor (AHR) has been the subject of intense study in the last decade, with its role in maintaining skin homeostasis brought under the spotlight, having previously been exclusively the focus of toxicological studies, owing to its activation by the toxin 2,3,7,8-tetrachlorodibenzo-p-dioxin (TCDD) as a result of occupational exposure [1] or, in extreme cases, targeted poisoning of political figures [2]. AHR’s function is now understood to be double-edged, with its physiological activation by endogenous ligands resulting in beneficial functions in the human body, for example, maintaining skin barrier integrity and gut homeostasis [3]. Adding to the growing evidence of AHR’s beneficial activation in inflammatory skin disease, the development of tapinarof, a topical AHR modulating drug recently approved for psoriasis and effective also for atopic dermatitis (AD), offers a novel therapeutic option for patients with mild-to-moderate psoriasis and AD, harnessing the anti-inflammatory function of AHR [4]. Here, we review the current literature, describing AHR functions in skin health and disease, focusing on psoriasis and AD where the AHR pathway has been more extensively investigated, highlighting how controlled activation of the pathway is being used in clinical settings to ameliorate skin inflammation and identifying discrepancies in the literature regarding AHR expression in psoriasis and AD that warrant further investigation.

## 2. The AHR Pathway

### AHR Activation

AHR has been extensively reviewed with various topics of focus [3,5,6,7,8]. AHR is an evolutionarily conserved ligand-activated transcription factor belonging to the basic Helix–Loop–Helix–Per-Arnt-Sim (bHLH-PAS) family and is located on chromosome 7p21.1. Prior to activation, it is held in an inactive state in the cytoplasm by associated chaperone proteins, 90kDa Heat Shock Protein (HSP90), p23, AHR-Interacting Protein (AIP/XAP2), and protein kinase SRC (Figure 1). HSP90 keeps AHR in a ligand-responsive conformation [9], whilst masking the DNA-binding element of the transcription factor. Upon ligand binding, this element is unmasked, allowing AHR to translocate into the nucleus where binding to its partner, the AHR Nuclear Translocator (ARNT), causes dissociation of the receptor from the chaperone complex [10] and subsequent binding to the DNA at regions possessing the Xenobiotic Response Element (XRE). This induces the transcription of the AHR gene battery, including the Cytochrome P450 (CYP) enzymes *CYP1A1*, *CYP1A2*, and *CYP1B1*, the AHR Repressor (*AHRR*), and TCDD-inducible poly(ADP-ribose) polymerase (*TIPARP*). The induction of *CYP1A1* expression is a bona fide biomarker of AHR activation, as this gene is solely controlled by AHR. The products of the AHR gene battery play a critical role in regulating the activity of AHR, each providing distinct negative feedback mechanisms, a strong indication of how important it is to keep the pathway in check. AHRR, identified first in mice [11] and then in humans [12] as a mechanism of AHR regulation, regulates AHR signalling by disrupting the AHR/ARNT dimer, preventing AHR-dependent gene transcription [13,14]. Ligated AHR, dissociated from chaperones, can be targeted for proteasomal degradation [15,16] via TiPARP [17], with TiPARP-KO mice resulting in elevated levels of AHR and increased sensitivity to TCDD-induced hepatotoxicity compared to wild-type [18]. Finally, CYP1s are xenobiotic metabolising enzymes and play a key role in regulating AHR activation by metabolising numerous AHR ligands, thus terminating signalling [19,20]. Importantly, CYP1 inhibitors, for example, 3′-methoxy-4′-nitroflavone (MNF), inhibit ligand metabolism, thereby promoting indirect AHR activation [21]. When AHR ligands are metabolised by CYP1s, they can either undergo bioactivation into toxic compounds, like the polycyclic aromatic hydrocarbon (PAH) Benzo[*a*]pyrene (BaP), which is oxidated into the carcinogen BaP-7,8-dihydrodiol-9,10-epoxide (BPDE) by CYP1A1 [22], or biotransformation into polar compounds, which can then be excreted, for example, 6-formylindolo[3,2-*b*]carbazole (FICZ), which is hydroxylated into excreted metabolites [23]. AHR ligands can be broadly categorised as physiological ligands and exogenous toxins, both of which can activate AHR for homeostatic and immunosuppressive effects. In addition to the ligand-induced (canonical) pathway of activation, non-canonical AHR signalling has also been reported. AHR interacts with other transcription factors to coordinate downstream gene activation, for example, epidermal growth factor receptor (EGFR) signalling, nuclear factor kappa-light-chain-enhancer of activated B cells (NF-κB), and nuclear factor erythroid 2-related factor 2 (NRF2). In skin, non-canonical AHR signalling has been reported in the context of the mutual interaction of activated AHR with the EGFR signalling pathway [24,25]. Non-canonical AHR activation, including its interaction with EGFR signalling in skin, has been recently reviewed in detail [26] and as such it will not be covered here. AHR has also been reported to interact with other receptors, including the vitamin D receptor (VDR) and liver X receptor (LXR). Expression of the VDR target gene *CYP24A1* was enhanced in human THP-1 cells by co-treatment with BaP, suggesting a mutual interaction between AHR and VDR [27]. A cross-talk between AHR and LXR was also previously reported [3], but a recent study has shown that simultaneous activation of LXR and AHR in zebrafish did not interfere with the expression of either receptors’ target genes [28]. Further investigations into AHR’s interactions with different receptors would be beneficial to fully elucidate the AHR signalling cascade.

## 3. AHR Ligands

### 3.1. Exogenous Ligands

First described in the 1970s as a receptor responsible for xenobiotic metabolism [29], AHR was thought to have exogenous ligands belonging to two classes: polyhalogenated aromatic hydrocarbons (HAHs) or dioxins, e.g., TCDD, and PAHs, e.g., BaP and 7,12-dimethylbenz(a)anthracene (DMBA) (Figure 2). TCDD is AHR’s most potent ligand, with its hydrophobic planar structure increasing its binding affinity [30]. TCDD is highly stable and resistant to CYP1 metabolism, leading to persistent signalling, which contributes to the reported liver toxicity and excessive immune suppression [31]. Dioxin exposure presents on the skin as chloracne, a chronic non-inflammatory skin condition characterised by epidermal hyperkeratinisation and cyst formation [32]. PAHs are chemical pollutants commonly found in the environment, with human exposure resulting from airborne particulates, such as cigarette smoke and exhaust fumes, or from the diet, such as combustion by-products generated upon cooking meat over open flames. PAHs are often carcinogenic following bioactivation by CYP1 enzymes, generating oxygenated reactive intermediates that can interact with DNA, initiating mutagenesis and carcinogenesis [33]. On the other hand, CYP1s detoxify PAH and other xenobiotics into polar derivatives excreted into the urine and bile [34] and it appears that the beneficial effects of CYP1-mediated detoxification are far more prominent than the detrimental effects due to xenobiotics bioactivation [35]. Moreover, a tightly controlled topical application of PAHs also appears to be beneficial, with the historical use of coal tar, a complex mixture of compounds including assorted PAHs [36] for the treatment of psoriasis [37] and AD [38]. Importantly, unlike most PAH exposures, coal tar is not associated with increased carcinogenesis [39].

### 3.2. Physiological Ligands and Activators

AHR is evolutionarily conserved, with homologs present in early vertebrates, nematodes, and drosophila [40,41,42]. The receptor has been retained through evolution as a metabolic adaptation to the exogenous ligands that have been present in the environment for millions of years [30]. However, its evolutionary conservation does indicate a physiological function for the receptor in humans, which has inferred the existence of physiological ligands for AHR. In this review, a physiological ligand is defined as a naturally derived, non-man-made compound capable of binding and activating AHR. In contrast to exogenous ligands, which are either metabolised into or undergo bioactivation into carcinogenic metabolites [43], physiological ligands undergo biotransformation into non-toxic metabolites for excretion [23], a distinction with important clinical implications in terms of safety for their therapeutic application. Early experiments performed in rats showed that vegetable dietary compounds in their chow induced CYP1 enzymatic activity, historically termed aryl hydrocarbon hydroxylase activity, suggesting AHR binding affinity [44,45]. The dietary compound indole-3-carbinol (I3C) has since been identified in cruciferous vegetables, including broccoli, cabbage, and Brussels sprouts, as the precursor of the high-affinity AHR ligand indolo[3,2-*b*]carbazole (ICZ) generated by the acidic conditions in the digestive tract [46]. A key source of physiological AHR ligands in skin are tryptophan metabolites [47], derived from tryptophan metabolism by the host microbiome, and from UV light exposure (Figure 2). Tryptophan is a precursor to many biologically relevant compounds, metabolised by three main pathways: kynurenine, serotonin, and indole pathways [48,49]. Approximately 95% of dietary tryptophan is used in the kynurenine pathway, with tryptophan metabolised into the compound kynurenine by indoleamine-2,3-dioxygenase (IDO) and tryptophan dioxygenase (TDO) [50]. While it has been suggested for some time that kynurenine is an AHR ligand with immunomodulatory effects [51,52], micromolar concentrations of kynurenine are required in vitro to elicit a response [53], compared to nanomolar concentrations for other physiological ligands like ICZ [46]. Instead, kynurenine may act as a pro-ligand, and its derivatives or Trace-Extended Aromatic Condensation Products (TEACOPs) are the actual AHR ligands, capable of eliciting a response at even picomolar concentrations [53]. The remaining dietary tryptophan can be converted into serotonin by tryptophan hydroxylase (TPH) 1 in the gut, and TPH2 in the brain, with implications in gastrointestinal homeostasis and neurological function [54,55]. The final metabolic pathway, and a key source of physiological AHR ligands in the skin and gut, is the indole pathway, which is active only in some components of the host microbiome, for example, *Lactobacilllus* [56] and *Malassezia* [57]. These microbes can metabolise tryptophan into indoles, which are associated with the maintenance of barrier homeostasis via AHR activation [58]. The endogenous ligand FICZ is another tryptophan derivative, produced by UV light [59,60]. FICZ is a highly potent AHR ligand [23] and is used as the prototypical activator of AHR in physiological studies, owing to its presence in vivo and its rapid metabolism by CYP1A1 [21]. Manuka honey, a dark honey only produced by bees feeding on the *Leptospermum scoparium* shrub in New Zealand and eastern Australia, has a complex formulation consisting of several biologically active substances, including indoles and flavonoids [61]. We have recently shown that manuka honey, and to a lesser extent other types of honey, can activate AHR both directly, possibly through indoles, and indirectly, through inhibition of CYP1 enzymatic activity, possibly by flavonoids [62]. Thus, manuka honey can be considered an AHR activator, due to the complex mixture of indoles as AHR ligands and flavonoids as CYP1 inhibitors. Finally, tapinarof, a naturally derived AHR activator produced by bacterial symbionts of nematodes, activates the AHR pathway to promote both anti-inflammatory and homeostatic functions in the skin [63], and has been approved for topical use in psoriasis [64]. This sheer variety of endogenous AHR ligands is strongly supporting of a physiological role for AHR, and the existence of the UV light-derived ligand, FICZ, anticipates a role for AHR in the skin, a UV light-exposed organ.

**Figure 2 ijms-26-01618-f002:**
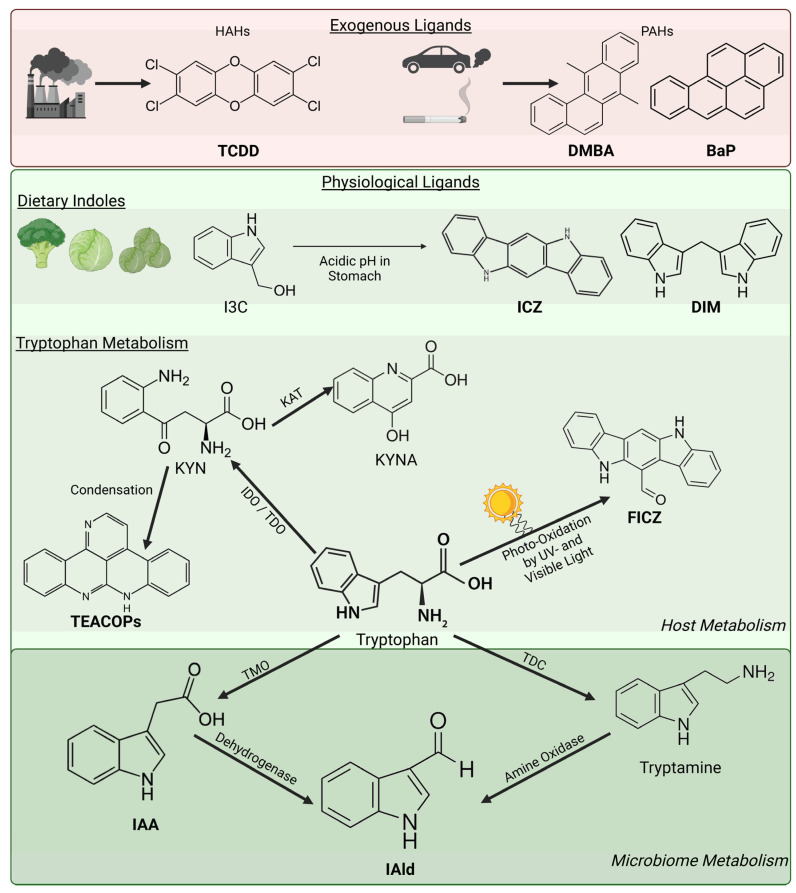
Examples of exogenous and physiological AHR ligands. AHR ligands can be derived from both exogenous and physiological sources. Side products of chemical manufacturing and air pollution from exhaust fumes and cigarette smoke are key sources of HAHs and PAHs capable of activating AHR. Physiological sources include the dietary compound I3C as a precursor of the high-affinity AHR ligands ICZ and DIM, as well as molecules derived from tryptophan metabolism by the host (light green; KYN, KYNA, TEACOPs, and FICZ) and host microbiome (dark green; IAA and IAld). AHR ligands are displayed with bold text. TEACOP structure from [53]. TCDD—2,3,7,8-tetrachlorodibenzo-p-dioxin, DMBA—7,12-dimethylbenz[*a*]anthracene, BaP—benzo[*a*]pyrene, I3C—indole-3-carbinol, ICZ—indolo[3,2-*b*]carbazole, DIM—3,3′-diindolylmethane, IDO—indoleamine-2,3-dioxygenase, TDO—tryptophan dioxygenase, KYN—kynurenine, KAT—kynurenine aminotransferase, KYNA—kynurenic acid, TEACOPs—Trace-Extended Aromatic Condensation Products, FICZ—6-formylindolo[3,2-*b*]carbazole, TMO—tryptophan monooxygenase, IAA—indole-3-acetic acid, IAld—indole-3-aldehyde, TDC—tryptophan decarboxylase. Created with biorender.com, 18 November 2024.

## 4. AHR Expression and Function in Healthy Skin

The skin is the most exposed organ to the external environment and provides a first line of defence against pathogens, environmental pollutants, and injury [65]. AHR is expressed in most cells of the skin, including keratinocytes, melanocytes, fibroblasts, and resident immune cells like Langerhans cells (LCs) and γδ T cells [3,66]. AHR is highly expressed in keratinocytes, where it is involved in keratinocyte differentiation and skin barrier formation [3]. Early studies using TCDD showed that AHR was involved in inducing terminal differentiation in keratinocytes [67], highlighting the importance of AHR in skin biology. AHR-deficient keratinocytes have a significant reduction in differentiation-specific gene expression, including reduced *Krt1*, *Lor*, and *Ivl*, with the reduction induced also by an AHR antagonist in wild-type keratinocytes [68], supporting AHR’s role in regulating epidermal homeostasis by promoting keratinocyte differentiation and skin barrier function. In addition, a recent study has shown that AHR directly targets the transcription factor TFAP2A to control keratinocyte differentiation [69], elucidating the mechanism by which AHR regulates the skin barrier. Activation of AHR through different means including coal tar [36], FICZ [70], manuka honey [62], and tapinarof [71] has been shown to upregulate *FLG* expression in keratinocytes. Young AHR-KO mice subjected to mechanical stress display elevated trans-epidermal water loss (TEWL) [72], a biomarker of skin barrier integrity, which is increased when the barrier is compromised, such as in inflammatory skin disease [73]. Indoles generated by the skin microbiome are critical for maintaining the skin barrier via AHR signalling in keratinocytes as shown in germ-free mice and keratinocyte-specific *Ahr* knockout mice (*K14^Cre^Ahr^f/f^*) [58]. Germ-free mice presented with an altered skin barrier, with downregulation in genes involved in cornified envelope formation, a defective keratinocyte differentiation programme, and increased TEWL upon mechanical stress, compared to conventionally raised mice, indicating barrier disruption. *K14^Cre^Ahr^f/f^* mice presented the same phenotype, highlighting the importance of the microbiome in regulating the barrier function via AHR signalling.

Most immune cells present in the skin also express AHR, including resident T helper (Th) 17 (Th17) cells, innate lymphoid cells (ILCs), and dendritic cells (DCs) [74,75]. AHR is involved in Th17 cell function [76]. AHR’s activation in Th17 cells [77] and ILC3s [20] is critical for Interleukin (IL)-22 production, which is involved in keratinocyte proliferation and differentiation. In contrast, AHR is marginally expressed in regulatory T (Treg) cells [77]; however, AHR’s exact role in Tregs remains controversial in the literature [78]. In the skin, one study suggested that AHR activation by a novel agonist GDU-952 resulted in amelioration of 2,4-dinitrofluorobenzene (DNFB)-induced inflammation in mice, with an increased frequency of CD4+FOXP3+ cells detected in the skin following GDU-952 treatment [79]. Further studies, both preclinical and clinical, will be required to fully elucidate AHR’s role in skin Tregs. In mice, AHR is absolutely required for the survival of epidermal Vγ3 T cells [80,81] (using Garman’s nomenclature [82], also called dendritic epidermal T cells (DETC) due to their morphology). DETCs are a resident immune cell population in the murine, but not human, epidermis, highly specialised in promoting skin homeostasis [83]. AHR is also required for the long-term persistence of tissue-resident memory CD8+ T cells (T_RM_) in the murine epidermis, as these cells are lost over time when AHR is knocked out [84]. AHR is involved in the maturation and tolerogenic function of LCs [85], promoting increased IL-10 production [86], as well as in LC development, given the reduced number of total LCs in mice where AHR has been conditionally knocked out in langerin-expressing cells (langerin-Ahr^−/−^ mice) [87]. Moreover, langerin-Ahr^−/−^ mice exposed to Ovalbumin (Ova) in an epicutaneous protein sensitisation model mounted an increased Th2 response [87], further supporting AHR’s anti-inflammatory role in LCs by reducing Th2 responses. How AHR’s role in LCs relates to the therapeutic activation of the pathway in humans remains to be elucidated. Finally, AHR has been shown to drive the differentiation of monocytes into monocyte-derived DCs (moDCs) instead of monocyte-derived macrophages, both in vitro using human monocytes and in vivo in the dermis of mice skin [88], and to inhibit the maturation of human moDCs in vitro [89]. Overall, AHR expression and function in a variety of skin immune cells support its anti-inflammatory and homeostatic role.

## 5. AHR in Inflammatory Skin Disease

### 5.1. AHR in Psoriasis

Psoriasis is a complex, chronic inflammatory skin disease affecting over 60 million people worldwide [90]. Disease etiopathogenesis results from the interaction of genetic susceptibility, environmental triggers, and immune dysregulation, which culminates in a pathogenic cross-talk between keratinocytes, DCs, and T cells releasing key cytokines TNF, IL-23, IL-17A, and IL-17F [91]. A critical role for AHR in psoriasis was uncovered in 2014, when we reported that a lack of AHR signalling exacerbates the severity of psoriasiform inflammation in mice, by unleashing excessive production of chemokines and cytokines by keratinocytes in response to proinflammatory stimuli [92]. This resulted in increased epidermal thickening, reduced keratinocyte differentiation, heightened skin neutrophilia, and increased expression of hallmark psoriasis cytokines *Tnf*, *Il23*, *Il17a*, and *Il17f* [92]. Conversely, AHR ligation by FICZ ameliorated the transcriptional profile of ex vivo skin biopsies from psoriasis patients and reduced the severity of psoriasiform inflammation in mice, further supporting a beneficial role of AHR in psoriasis. Following on from this finding, AHR modulators have been investigated as potential therapeutics for inflammatory skin diseases. Preclinical studies of tapinarof showed significant improvement in the psoriasis-like phenotype induced by imiquimod in mice with a reduction in Th17 cytokines and reduced epidermal thickness [71]. Further investigation into tapinarof’s mechanism of action uncovered both anti-inflammatory and antioxidant mechanisms in psoriasis skin [93]. Moreover, a published abstract from a conference has suggested that tapinarof and TCDD do not compete for binding to AHR, likely engaging via distinct binding pockets on the receptor [93,94]; further studies should be undertaken to confirm this finding and how it relates to tapinarof’s mechanism of action. The results from four phase 1 trials to investigate the safety of topical tapinarof showed that it was well tolerated, with no degree of skin sensitisation or photo-allergenicity; however, a slight potential for mild cumulative irritation was reported [95]. Following a successful phase 2b dose-finding trial in psoriasis [96,97], two phase 3 trials (PSOARING 1 and PSOARING 2) investigated the clinical efficacy and safety of 1% topical tapinarof cream in 683 adults with mild-to-severe plaque psoriasis [64]. Disease severity was assessed by the 5-point scale Physician’s Global Assessment (PGA), with a score of 2, 3, or 4 relating to mild, moderate, or severe psoriasis, respectively, as well as using the Psoriasis Area and Severity Index (PASI), evaluating the body surface area involvement, erythema, induration, and scaling of psoriasis lesions [98]. At week 12, up to 40% of psoriasis patients applying tapinarof achieved the primary endpoint of PGA response of 0 (clear) or 1 (almost clear), with a decrease from the patient’s baseline PGA by at least 2 points, compared to up to 6% in the vehicle. Up to 47% of patients met the secondary endpoint of PASI-75, which is a 75% reduction in baseline disease severity, compared to up to 10% in the vehicle group [64]. The results from the long-term extension trial, PSOARING 3, which assessed the long-term efficacy of tapinarof, showed that clinical response was maintained for at least 4 months after cessation of treatment [99]. Adverse events associated with tapinarof included folliculitis, headache, nasopharyngitis, upper respiratory tract infection, and contact dermatitis. Local irritation due to tapinarof cream was rated as no worse than mild by patients and investigators, even when applied to sensitive skin. In May 2022, 1% tapinarof was FDA-approved for mild-to-moderate adult plaque psoriasis [100], making it the first-in-class AHR modulating drug, as well as the first novel topical treatment for psoriasis developed in 25 years [100]. Benvitimod, approved in China following a phase 3 clinical trial [101], shares the same active ingredient with tapinarof, but has a different formulation and dose regime, meaning the drugs cannot be directly compared [102].

The clinical efficacy of tapinarof lends further support to the therapeutic targeting of the AHR pathway to harness its homeostatic and anti-inflammatory effect in skin inflammation. The existence of critical checkpoints to control AHR activation offers opportunities for alternative approaches to promote its beneficial effects, beyond ligands’ supplementation [103]. We have focused on the negative feedback provided by CYP1A1 to control AHR activation via ligand metabolism and shown that *Cyp1a1* overexpression and excessive metabolic activity results in more severe psoriasis-like skin inflammation in mice [104], a direct phenocopy of the exacerbated pathology observed in mice lacking *Ahr* [92], possibly due to reduced ligand availability, as shown in the gut [20]. Thus, a tightly controlled AHR/CYP1A1 axis is required to maintain skin homeostasis and avoid inflammation. Interestingly, we found excessive CYP1A1 enzymatic activity in in vitro-expanded Th17 cells from psoriasis patients [104], suggesting a possible dysregulation of the AHR/CYP1A1 axis in psoriasis leading to reduced AHR activation. Indeed, data about AHR expression and activation in psoriasis compared to non-lesional or healthy skin are sometimes conflicting (Table 1).

Three studies report increased AHR transcript and protein expression in psoriasis lesional versus healthy skin [105,106,107]. Another study analysing publicly available microarray data [108] also reports increased *AHR* mRNA expression in psoriasis lesional compared to matched psoriasis perilesional skin [109], but there is no direct comparison with healthy skin. However, the same study reports decreased AHR protein expression in lesional versus healthy skin in their own small independent cohorts of psoriasis patients and healthy controls [109]. We have also detected decreased *AHR* mRNA expression in psoriasis lesional as compared to healthy skin, as well as in whole blood, in our ongoing investigation of the AHR pathway in skin inflammation [110]. How to reconcile these discrepancies and how these findings relate to the beneficial role of AHR activation in psoriasis remains to be clarified. Moreover, the recent identification of psoriasis susceptibility SNPs near the *AHR* locus [111] not only lends further support to the importance of AHR in psoriasis pathogenesis but also raises the possibility of genetically driven influences on AHR expression.

On the other hand, *CYP1A1* mRNA expression, a biomarker of AHR activation, is reported to be increased in lesional psoriasis compared to healthy skin in two small studies performed in Korea [105,106], while it is decreased in Zhu et al.’s [109] and in our own study [104], both performed in predominantly White cohorts, as well as in another study performed in Korea (Table 1). Importantly, differential expression of AHR between studies may suggest inter-individual variability in response to tapinarof because of higher or lower AHR expression in these patients. Real-world data on tapinarof’s use in the clinic will be crucial to understand whether this is the case. The different genetic background of the populations studied, as well as environmental factors, may explain at least some of the discrepancies observed. For example, the reported dysbiosis of the psoriasis skin microbiome, with a reduction in certain commensals such as *Lactobacilli* [112], may be responsible for reduced availability of AHR ligands, resulting in reduced AHR activation and CYP1A1 expression. Nevertheless, as CYP1A1 is at the same time a biomarker of AHR activation and a critical checkpoint controlling AHR signalling [21,104], it will be important to clarify these discrepancies and fully dissect the AHR/CYP1A1 axis to identify dysregulated checkpoints amenable for therapeutic intervention.

**Table 1 ijms-26-01618-t001:** AHR and CYP1A1 expression in psoriasis.

Study	Tissue	Ethnicity	Country of Study	Cohort	Biomolecule/ Technique	AHR Expression	CYP1A1 Expression
Kim et al., 2014 [105]	Skin	N/R	Korea	15 PsL vs. 16 HV	mRNA/ RT-PCR	**↑** PsL (*p* < 0.001)	**↓** PsL (*p* < 0.001)
				11 PsL vs. 7 HV	Protein/ IHC	**↑** PsL (*p* < 0.001)	N/A
Kim et al., 2020 [106]	Skin	N/R	Korea	3 PsL vs. 3 HV	mRNA/ qRT-PCR	**↑** PsL (*p* < 0.05)	**↑** PsL (*p* < 0.05)
				3 PsL vs. 3 HV	Protein/ IHC	**↑** PsL	**↑** PsL
Kim et al., 2021 [107]	Skin	N/R	Korea	5 PsL vs. 5 Hv	mRNA/ qRT-PCR	**↑** PsL (*p* < 0.01)	**↑** PsL (*p* < 0.001)
				24 PsL vs. 10 HV	Protein/ IHC	**↑** PsL (*p* < 0.001)	**↑** PsL (*p* < 0.001)
				5 PsL vs. 5 HV	Protein/ IF	**↑** PsL (*p* < 0.001)	N/A
Kyoreva et al., 2021 [104]	Skin	90.8% Caucasian	UK	22 PsL/NL vs. 42 HV	mRNA/ qRT-PCR	N/A	**↓** PsL (*p* < 0.01)
Zhu et al., 2020 [109]	Skin	84.3% Caucasian	USA	85 PsL vs. Ps-PeriL	mRNA/ Microarray [108]	**↑** PsL (*p* = 0.02)	**↓** PsL (*p* < 0.0001)
		N/R	China	4 PsL vs. 4 HV	Protein/ Immunoblot	**↓** PsL (*p* < 0.05)	N/A
				3 PsL vs. 3 HV	Protein/ IHC	**↓** PsL	N/A

*p* Values where available; RT-PCR: Real-Time Polymerase Chain Reaction, qRT-PCR: Quantitative Reverse-Transcription PCR, IHC: Immunohistochemistry, IF: Immunofluorescence, Ps: Psoriasis, PsL: Lesional Psoriasis, Ps-PeriL: Perilesional Psoriasis, HV: Healthy Volunteer, N/R: Not Reported, N/A: Not Applicable, ↑: Increased, ↓: Decreased.

### 5.2. AHR in Atopic Dermatitis

AD is a chronic inflammatory skin disease, affecting over 223 million people worldwide [113]. Four main factors contribute to AD pathology, with (i) genetic susceptibility, (ii) immune dysregulation, and (iii) environmental factors all contributing to the (iv) epidermal barrier dysfunction. AD skin is also characterised by a severely dysbiotic microbiome, highly dominated by pathogenic bacteria like *Staphylococcus* and a reduction in commensal microorganisms such as *Lactobacilli* [112]. Impairment of the skin barrier, either intrinsic (i.e., due to mutation in *FLG*) or extrinsic (i.e., due to scratching), is a key driver for AD initiation and further exacerbates the immune dysregulation, resulting in a vicious cycle of heightened inflammation and barrier disruption [114]. Termed the *Goeckerman regimen* after William Goeckerman who developed the method [115], coal tar has been used for many years as topical therapy for AD in conjunction with phototherapy, resulting in significant amelioration of AD symptoms for over 7 months [38,116]. The mechanism of action of coal tar was eventually elucidated in a pivotal study by van den Bogaard and coworkers, who showed that topical coal tar treatment restores the expression of barrier proteins like filaggrin and involucrin in an AHR-dependent manner [36]. Moreover, coal tar treatment was able to revert Th2-induced AD hallmarks, such as spongiosis and increased *CCL26* expression, in a skin-equivalent experimental model [36]. Coal tar also modulates the AD skin microbiome by reducing the abundance of pathogenic bacteria like *Staphylococcus* and increasing the expression of antimicrobial peptides like LL-37 and LCE3A in an AHR-dependent manner [117].

On the other hand, AHR has been linked to AD pathogenesis when chronically activated by exogenous ligands. Hidaka et.al. linked AHR activation to AD via the pruritogenic factor artemin [118], which is increased in AD, and its production is directly related to AHR activation by DMBA, a PAH. Chronic applications of DMBA on mice skin resulted in an AD-like phenotype, with increased inflammation and epidermal hyperplasia [118]. This AD-like phenotype has also been shown in a constitutively activated AHR (AHR-CA) mouse model [118,119], where AHR expression is controlled by the keratin 5 promoter (*Krt5^Cre^Ahr^fl/fl^* mice), localising the AHR-CA to the keratinocytes of the basal layer. *Krt5^Cre^Ahr^fl/fl^* mice experience excessive skin inflammation reminiscent of AD immunopathology with increased *Il4*, *Il13*, and *Tslp*, hyperkeratosis of the skin, and increased pruritis, all phenotypes of AD. While these studies appear to suggest a role for AHR in promoting AD pathogenesis, they have used chronic applications of exogenous ligands like PAHs or uncontrolled AHR activation. Within the same study, Hidaka et.al. showed that the chronic application of FICZ, a physiological ligand, did not generate the same inflammatory phenotype present following DMBA application, and FICZ stimulation did not induce artemin expression [118]. Indeed, in chronic mite-induced dermatitis, a mouse model of AD-like inflammation, FICZ treatment results in the amelioration of inflammation and restores barrier function by increasing *Flg* expression [70]. Moreover, physiological activation of AHR reduces *TSLP* expression in human keratinocytes [120]. Thus, while uncontrolled or chronic AHR activation in the skin may have a detrimental effect, physiological or controlled AHR activation not only appears devoid of the same negative effects, but it restores a compromised skin barrier and reduces inflammation in AD. Indeed, following on from the success of the psoriasis trials, tapinarof has been investigated as a therapy for AD. In a phase 1 trial investigating safety and pharmacokinetics, tapinarof was systemically absorbed at minimal concentrations, as shown by the measurable detection of the drug in patients’ plasma, with higher concentrations detected in the 2% group compared to 1% tapinarof [121]. Intriguingly, the mean concentration detected was lower towards the end of the trial, suggesting a lack of accumulation in the body and an improved skin barrier [121]. Most importantly, 1% tapinarof was well tolerated, with no severe adverse effects reported. Two phase 3 trials (ADORING 1 and ADORING 2) in the USA investigated the clinical efficacy and safety of 1% tapinarof cream in 541 adults and children with moderate-to-severe AD [122]. Disease severity was assessed using the Validated Investigator Global Assessment for Atopic Dermatitis (vIGA-AD), with a score of 3 or 4 relating to moderate or severe AD, respectively, and using the Eczema Area and Severity Index (EASI), which considers the area of involvement and lesion intensity, assessing erythema, oedema, excoriation, and lichenification [123]. After 8 weeks, up to 46% of patients applying tapinarof achieved the primary endpoint of a vIGA-AD response of 0 (clear) or 1 (almost clear), with a decrease from the patient’s baseline vIGA-AD by at least 2 points, compared to up to 18% in the vehicle group. Moreover, up to 59% of patients in the active treatment arm achieved the secondary endpoint of EASI-75, which is a 75% reduction in the baseline EASI, compared to up to 22% in the vehicle group. Consistent with the PSOARING trials, the common adverse events reported included headache, nasopharyngitis, and contact dermatitis, with up to 0.7% of patients discontinuing the treatment. Patients from ADORING 1 and 2 could enrol in the long-term extension trial, ADORING 3, which has assessed the long-term efficacy of tapinarof in AD, with the final results pending as of November 2024 [124].

Importantly, the AHR pathway appears dysregulated in AD, based on differential AHR and CYP1A1 expression in the skin and blood of AD patients compared to healthy controls (Table 2). AHR transcript and protein expression is consistently increased in AD lesional skin as well as in AD keratinocytes and PBMCs compared to healthy. On the contrary, reports on CYP1A1 expression are conflicting, with studies performed in Asia [105,118,125] reporting an increase in *CYP1A1* mRNA in AD lesional skin and PBMCs while a study [126] analysing publicly available datasets generated in Western countries [127,128] reports decreased *CYP1A1* in AD skin and keratinocytes versus healthy. As for psoriasis, genetic and/or environmental factors may be responsible for these discrepancies. For instance, different pollution levels at the locations of each study may be implicated, as CYP1A1 can be induced by PAHs, which are prevalent in heavily polluted areas. A positive correlation between *AHR* expression in the skin and AD severity has also been reported [125,129]. Nevertheless, a dysregulation in the AHR pathway is clear in AD, with increased AHR expression but possibly reduced activation and failure to constrain inflammation and maintain skin homeostasis. A dysregulation in tryptophan metabolism in the AD microbiome has also been reported [130], potentially generating fewer AHR ligands on the skin. For example, indole-3-aldehyde, an AHR-activating compound generated by tryptophan metabolism, is significantly decreased in the skin of AD patients compared to healthy [131]. As for psoriasis, a deep investigation of the AHR pathway in AD is needed to clarify the current discrepancies, better understand the complex regulation of the pathway, and potentially identify novel therapeutic targets.

A summary of the current knowledge on the AHR pathway in the two inflammatory skin diseases where it has been studied most extensively, i.e., psoriasis and atopic dermatitis, versus healthy skin is displayed in Figure 3.

**Table 2 ijms-26-01618-t002:** AHR and CYP1A1 expression in AD.

Study	Tissue	Ethnicity	Country of Study	Cohort	Biomolecule/Technique	AHR Expression	CYP1A1 Expression
Kim et al., 2014 [105]	Skin	N/R	Korea	19 ADL vs. 22 HV	mRNA/ RT-PCR	Trend **↑** ADL	**↑** ADL (*p* < 0.001)
				17 ADL vs. 7 HV	Protein/ IHC	**↑** ADL	N/A
Hong et al., 2016 [132]	Skin	N/R	Taiwan	8 ADL vs. 6 HV	Protein/ IF	**↑** ADL (*p* < 0.05)	No Difference
Hidaka et al., 2017 [118]	Skin	N/R	Japan	20 ADL vs. 20 HV	mRNA/ RNAscope	N/A	**↑** ADL (*p* < 0.01)
				3 ADL vs. 3 HV	Protein/ IHC	N/A	**↑** ADL
Hu et al., 2020 [125]	PBMCs	N/R	China	29 ADL vs. 17 HV	mRNA/ RT-PCR	**↑** ADL (*p* < 0.05)	**↑** ADL (*p* < 0.05)
	Skin	N/R	China	ADL vs. HV#	Protein/ IHC	**↑** ADL (*p* < 0.001)	N/A
Hu and Zhang, 2023 [133]	PBMCs	N/R	China	20 ADL vs. 18 HV	mRNA/ qRT-PCR	**↑** ADL (*p* < 0.05)	N/A
Proper et al., 2024 [126]	Skin	N/R	Germany	21 ADL vs. 38 HV	mRNA/ RNAseq [127]	Trend **↑** ADL (*p* = 0.056)	**↓** ADL (*p* < 0.0001)
		N/R	USA	5 ADL vs. 6 HV	mRNA/ scRNAseq [128]	**↑** ADL KC (*p* < 0.0001)	**↓** ADL KC (*p* < 0.01)

PBMCs: Peripheral Blood Mononuclear Cells, KC: Keratinocyte, RT-PCR: Real-Time Polymerase Chain Reaction, qRT-PCR: Quantitative Reverse-Transcription PCR, RNAseq: RNA Sequencing, scRNAseq: Single-Cell RNAseq, IHC: Immunohistochemistry, IF: Immunofluorescence, AD: Atopic Dermatitis, HV: Healthy Volunteer, N/R: Not Reported, N/A: Not Applicable, ↑: Increased, ↓: Decreased, #: No Sample Size Reported.

## 6. Conclusions and Perspectives

The existence of physiological ligands for AHR postulated a homeostatic role for this environmental sensor, widely expressed throughout the human body and particularly at barrier sites, such as the skin. Notwithstanding the toxicity induced by man-made toxins and environmental pollutants that has prevented an unbiased appreciation of AHR’s homeostatic role for a long time, the clinical efficacy of tapinarof in trials of psoriasis and AD firmly highlights the beneficial role of controlled AHR activation in human skin. These trials have validated AHR as a therapeutic target for inflammatory skin disease and provide a rationale for further investigation into its use in other inflammatory skin conditions [4]. For example, a dysregulation in tryptophan metabolism and a reduction in AHR activation, shown as reduced expression of *CYP1A1*, *AHRR*, and *CYP1A2,* has been detected in the skin of patients with hidradenitis suppurativa (HS) [134]. Moreover, case reports of efficacious off-label use of tapinarof in vitiligo [135], seborrheic dermatitis [136,137], and palmoplantar keratoderma [138] provide further evidence for controlled AHR modulation as a therapeutic strategy for skin disease, warranting further investigation in clinical trials. Other inflammatory skin diseases like contact dermatitis, acne, and frontal fibrosing alopecia (FFA) may also benefit from AHR modulation as a possible therapeutic strategy. Hapten-induced contact hypersensitivity (CHS) is an established mouse model of allergic contact dermatitis where the skin is sensitised to haptens like DNFB or FITC applied to the dorsal skin, followed by hapten exposure on the ear five days later [139]. A potential role for AHR in contact dermatitis has been explored using these models, with AHR^−/−^ mice experiencing a reduced CHS response compared to wild-type [85] using FITC-induced CHS. Conversely, using a DNFB-induced CHS model, indirect AHR activation using 4-n-nonylphenol (NP) also reduced the CHS response [140]. The only human study of contact dermatitis conducted in 1982 reported reduced activity of aryl hydrocarbon hydroxylase (the historical term for CYP1A1) in both irritant and allergic contact dermatitis patients [141]. Indoles have been shown to attenuate inflammation in an AHR-dependent manner in a rat model of acne vulgaris [142], a chronic inflammatory disease of the pilosebaceous unit of the skin [143], not to be confused with chloracne, which is the non-inflammatory skin condition resulting from dioxin exposure [32]. Finally, the identification of a putative causal missense variant in *CYP1B1* for FFA [144] provides genetic evidence for an involvement of the AHR pathway, suggesting that AHR modulation may be a useful therapeutic strategy for FFA. Finally, reconciling some of the current discrepancies reported in psoriasis and AD in terms of AHR expression and activation may aid in identifying additional checkpoints within the pathway that are amenable for therapeutic intervention in skin inflammation.

## Figures and Tables

**Figure 1 ijms-26-01618-f001:**
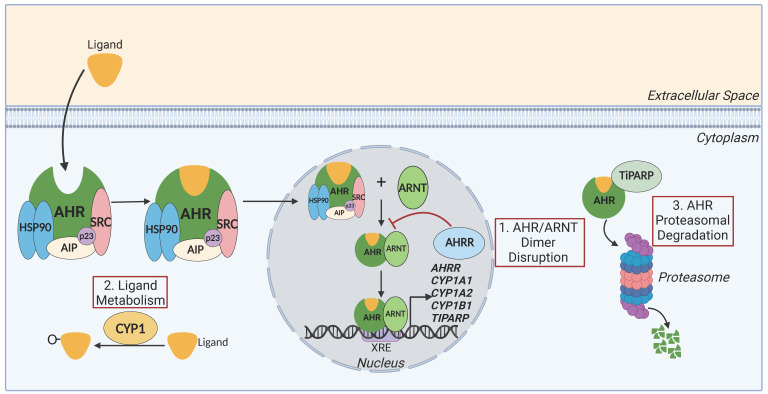
The AHR pathway. AHR is held in an inactive state in the cytoplasm by chaperone proteins HSP90, AIP, p23, and protein kinase SRC, preventing its degradation, and keeping it in a high-affinity conformation for ligands. Once bound by a ligand, AHR and its chaperone complex can translocate into the nucleus, where the ARNT can bind to AHR, causing its dissociation from the complex. The AHR/ARNT dimer can then bind to regions of DNA that possess the XRE (purple box), initiating transcription of AHR’s target genes *AHRR*, *CYP1A1*, *CYP1A2*, *CYP1B1*, and *TIPARP*. Following activation, AHR is highly regulated at three levels (red boxes). To prevent overactivation of the pathway, (1) the AHR/ARNT dimer can be disrupted by the AHRR binding to ARNT, (2) ligands can be metabolised by CYP1 enzymes by oxidation, and (3) AHR can be degraded by the proteasome via TiPARP. AHR—aryl hydrocarbon receptor, HSP90—Heat Shock Protein 90, AIP—AHR-Interacting Protein, ARNT—AHR Nuclear Translocator, XRE—Xenobiotic Response Element, CYP1—Cytochrome P450 1, AHRR—AHR Repressor, TiPARP—TCDD-inducible poly(ADP-ribose) polymerase. Created with biorender.com, 18 November 2024.

**Figure 3 ijms-26-01618-f003:**
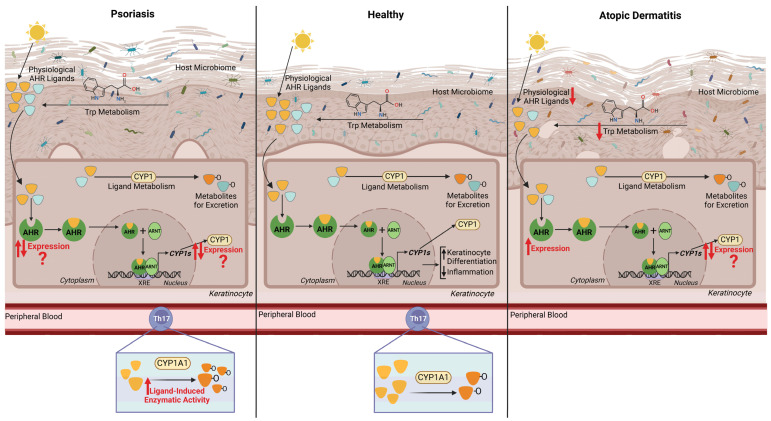
Summary of the current literature on the AHR pathway in psoriasis and atopic dermatitis. AHR is activated by physiological ligands derived from the host microbiome (light blue) and UV and visible light (yellow). The full pathway is described in Figure 1. In healthy skin (**middle panel**), AHR induces keratinocyte differentiation (black arrow upwards) and reduces inflammation (black arrow downwards) to maintain skin barrier homeostasis. The host microbiome is displayed in shades of blue representing healthy commensal microorganisms. For psoriasis lesional skin (**left panel**), the literature reports both increased and decreased AHR and CYP1A1 expression compared to healthy (red arrows for increased and decreased expression, and red question marks reflecting reported discrepancies). Ligand-induced CYP1 activity is increased in blood-derived Th17 cells of psoriasis patients. The host microbiome is displayed to reflect a dysbiosis from healthy, with a reduction in commensal microorganisms. In atopic dermatitis (**right panel**), increased AHR expression has been consistently reported (red arrow upwards); however, CYP1A1 expression is conflicting in the literature (red arrows for increased and decreased expression). The host microbiome is displayed in multiple colours to reflect a high degree of dysbiosis in AD with the presence of pathogenic bacteria (reds and browns). Tryptophan metabolism by the host microbiome is decreased, with a reduction in an AHR ligand produced as a consequence. AHR—aryl hydrocarbon receptor, ARNT—AHR Nuclear Translocator, XRE—Xenobiotic Response Element, CYP1—Cytochrome P450 1, Trp—tryptophan. Created with biorender.com, 18 November 2024.

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
