# Peer review of "The Aryl Hydrocarbon Receptor (AHR): Peacekeeper of the Skin"

_ijms, 2025, doi:10.3390/ijms26041618_

Round 1
Reviewer 1 Report
Comments and Suggestions for Authors
The paper focuses on the Aryl Hydrocarbon Receptor (AHR) and addresses a review of the literature on AHR and skin.
After an initial description of the receptor and setting up ligands and functions, the review focuses on the role of AHR on skin with psoriasis and atopic dermatitis.
The paper is very well done, straightforward and very orderly in its exposition.
It is clear and exposes the topic with a considerable level of detail, which is useful for both clinicians and molecular biologists.
The figures are clear and elegant. The tables and legends help and support the understanding of the text.
Bibliography is very comprehensive.
English is fluent.
All in all, this is an excellent, comprehensive and clear work that can be a reference and a base on the subject of AHR and skin.
Author Response
Comment:
The paper focuses on the Aryl Hydrocarbon Receptor (AHR) and addresses a review of the literature on AHR and skin.
After an initial description of the receptor and setting up ligands and functions, the review focuses on the role of AHR on skin with psoriasis and atopic dermatitis.
The paper is very well done, straightforward and very orderly in its exposition.
It is clear and exposes the topic with a considerable level of detail, which is useful for both clinicians and molecular biologists.
The figures are clear and elegant. The tables and legends help and support the understanding of the text.
Bibliography is very comprehensive.
English is fluent.
All in all, this is an excellent, comprehensive and clear work that can be a reference and a base on the subject of AHR and skin.
Response:
We thank Reviewer 1 for their positive comments about our review article.
Reviewer 2 Report
Comments and Suggestions for Authors
The authors present a review of the role of AHR in the skin homeostasis and disease. Although they did a good job at summarizing some of the most relevant findings, there are some important issues that need to be addressed before accepting the manuscript.
1) The review, although relevant, is similar in topic and structure to a previously published review in MDPI, 3 years ago (10.3390/cells10123559), please make sure the manuscript provides updated information compared to that review. Although the references seem updated, please verify this to avoid any ethical concerns.
2) AhR has interactions with other receptors like VDRs and LXRs. That interaction might play a role in skin diseases and homeostasis. Please add some information about that.
3) Can you please add any information about AhR and Contact Dermatitis? It will enhance the understanding of AhR in skin and its role in modulating Th1/Th2/Th17 responses. Also, please add a paragraph summarizing the role of AhR in Tregs, either in healthy or disease contexts.
4) In other reviews, people have addressed Acne and HS as part of the study of AhR in skin as a homeostatic and disease regulator. Can you please include this?
5) The summarized studies seem to have analyzed mRNA/Cyp1A1 expression using simple mRNA (RT PCR) or microarray technology. Are there any studies analyzing the expression of these genes at the single cell level in any disease (scRNA seq)? I am asking this because Ahr is greatly expressed in cells of hematopoietic origin and understanding the real cell type that is expressing these genes is of great relevance.
6) I suggest a modification in figure 3. Include the Th2 label in atopic dermatitis (like the Th17), and if possible, add other skin diseases. Also, in figure 1 make sure to place the legend about Biorender at the end of legend (after abbreviations)
Please work on these comments.
Author Response
Comment 1:
The review, although relevant, is similar in topic and structure to a previously published review in MDPI, 3 years ago (10.3390/cells10123559), please make sure the manuscript provides updated information compared to that review. Although the references seem updated, please verify this to avoid any ethical concerns.
Response 1:
In light of this comment, we have carefully compared our review article to the one by Napolitano et al mentioned here. We note that only 16 out of 144 articles cited and discussed in our review were also cited and discussed by Napolitano et al. Thus, we discuss different sub-topics related to AHR in inflammatory skin disease. For instance, in our manuscript we extensively discuss the current literature on AHR and CYP1A1 expression highlighting the discrepancies between the studies. Moreover, we also discuss the latest results of the ADORING trials for Tapinarof in atopic dermatitis which were not available at the time of the Napolitano publication. Therefore, we believe our review article is significantly different from the one by Napolitano et al and provides a more recent critical review of the field.
Comment 2:
AhR has interactions with other receptors like VDRs and LXRs. That interaction might play a role in skin diseases and homeostasis. Please add some information about that.
Response 2:
We thank the reviewer for this comment. We have added a short paragraph describing AHR’s interaction with other receptors (page 2, line 88-95).
“AHR has also been reported to interact with other receptors, including the vitamin D receptor (VDR) and liver X receptor (LXR). Expression of the VDR target gene CYP24A1 was enhanced in human THP-1 cells by co-treatment with BaP, suggesting a mutual interaction between AHR and VDR [27]. A cross-talk between AHR and LXR was also previously reported [3], but a recent study has shown that simultaneous activation of LXR and AHR in zebrafish did not interfere with the expression of either receptors’ target genes [28]. Further investigations into AHR’s interactions with different receptors would be beneficial to fully elucidate the AHR signaling cascade.”
Comment 3:
Can you please add any information about AhR and Contact Dermatitis? It will enhance the understanding of AhR in skin and its role in modulating Th1/Th2/Th17 responses. Also, please add a paragraph summarizing the role of AhR in Tregs, either in healthy or disease contexts.
Response 3:
We have included contact dermatitis among the skin conditions where AHR appears to have a role, based on the literature available. We report that murine models of allergic contact dermatitis highlight a potential role for AHR within the disease, and briefly mention the only human study, conducted in 1982 by Singh et al, reporting reduced activity of aryl hydrocarbon hydroxylase, the historical term for CYP1A1, in both irritant and allergic contact dermatitis patients (page 13, lines 486-497).
“Other inflammatory skin diseases like contact dermatitis, acne, and frontal fibrosing alopecia (FFA) may also benefit from AHR modulation as a possible therapeutic strategy. Hapten-induced contact hypersensitivity (CHS) is an established mouse model of allergic contact dermatitis where the skin is sensitised to haptens like DNFB or FITC applied to the dorsal skin, followed by hapten exposure on the ear five days later [139]. A potential role for AHR in contact dermatitis has been explored using these models, with AHR-/- mice experiencing a reduced CHS response compared to wild-type [85] using FITC-induced CHS. Conversely, using a DNFB-induced CHS model, indirect AHR activation using 4-n-nonylphenol (NP) also reduced the CHS response [140]. The only human study of contact dermatitis conducted in 1982 reported reduced activity of aryl hydrocarbon hydroxylase (the historical term for CYP1A1) in both irritant and allergic contact dermatitis patients [141].”
With regards to Tregs, we have briefly touched on the controversies in the literature about AHR role in Tregs, given its marginal expression in this cell type. As this review article is focused on skin, we briefly describe the only study reporting on a possible role for AHR in skin Tregs in a DNFB-induced model of skin inflammation (page 6, lines 233-239).
“In contrast, AHR is marginally expressed in regulatory T (Treg) cells [77], however AHR’s exact role in Tregs remains controversial in the literature [78]. In skin, one study suggested that AHR activation by a novel agonist GDU-952 resulted in amelioration of 2,4-dinitrofluorobenzene (DNFB)-induced inflammation in mice, with an increased frequency of CD4+FOXP3+ cells detected in the skin following GDU-952 treatment [79]. Further studies, both preclinical and clinical, will be required to fully elucidate AHR’s role in skin Tregs.”
Comment 4:
In other reviews, people have addressed Acne and HS as part of the study of AhR in skin as a homeostatic and disease regulator. Can you please include this?
Response 4:
The manuscript already included HS as one of the skin conditions where early data available suggest an involvement of AHR (page 13, lines 480-482) and briefly mentioned the only study investigating the AHR pathway in HS showing dysregulation in tryptophan metabolism and impaired AHR activation (Guenin-Mace et.al., 2020, JCI Insight, Ref 134).
Moreover, to address this and comment 3, we now also include information on contact dermatitis, acne, as well as frontal fibrosing alopecia (FFA), as additional examples of inflammatory skin diseases where evidence in the literature support a role for the AHR pathway and its therapeutic modulation (page 13, lines 486-503).
“Other inflammatory skin diseases like contact dermatitis, acne, and frontal fibrosing alopecia (FFA) may also benefit from AHR modulation as a possible therapeutic strategy. Hapten-induced contact hypersensitivity (CHS) is an established mouse model of allergic contact dermatitis where the skin is sensitised to haptens like DNFB or FITC applied to the dorsal skin, followed by hapten exposure on the ear five days later [139]. A potential role for AHR in contact dermatitis has been explored using these models, with AHR-/- mice experiencing a reduced CHS response compared to wild-type [85] using FITC-induced CHS. Conversely, using a DNFB-induced CHS model, indirect AHR activation using 4-n-nonylphenol (NP) also reduced the CHS response [140]. The only human study of contact dermatitis conducted in 1982 reported reduced activity of aryl hydrocarbon hydroxylase (the historical term for CYP1A1) in both irritant and allergic contact dermatitis patients [141]. Indoles have been shown to attenuate inflammation in an AHR-dependent manner in a rat model of acne vulgaris [142], a chronic inflammatory disease of the pilosebaceous unit of the skin [143], not to be confused with chloracne, which is the non-inflammatory skin condition resulting from dioxin exposure [32]. Finally, the identification of a putative causal missense variant in CYP1B1 for FFA [144] provides genetic evidence for an involvement of the AHR pathway, suggesting that AHR modulation may be a useful therapeutic strategy for FFA.”
Comment 5:
The summarized studies seem to have analyzed mRNA/Cyp1A1 expression using simple mRNA (RT PCR) or microarray technology. Are there any studies analyzing the expression of these genes at the single cell level in any disease (scRNA seq)? I am asking this because Ahr is greatly expressed in cells of hematopoietic origin and understanding the real cell type that is expressing these genes is of great relevance.
Response 5:
Table 2 (page 11, line 448) already included the study by Proper et.al., 2024 which specifically report AHR and CYP1A1 expression analysed in two publicly available dataset of AD samples (Tsoi et.al., 2019 for RNAseq, and He et.al., 2020 for scRNAseq). To the best of our knowledge, this is the only study reporting specifically scRNAseq data for AHR and CYP1A1 in psoriasis or AD skin compared to healthy skin.
We are aware of the study by Jin et.al., 2024 which performs scRNAseq on PBMCs from AD patients and healthy controls and reported high expression of AHR in cell clusters identified to be CD4_Effector_T, CD4_Memory_T, Prolif_Lym, and ILC2s, we have now added the citation when discussing which cells express AHR (Ref 75), but as there is no comparison between AD and healthy controls the study is not included in Table 2.
Comment 6a:
I suggest a modification in figure 3. Include the Th2 label in atopic dermatitis (like the Th17), and if possible, add other skin diseases.
Response 6a:
We display Th17 cells in the figure as published data (Kyoreva et al., 2021; Ref 104) show increased CYP1 activity in Th17 from psoriasis patients compared to healthy. No study has looked at CYP1 activity in AD so we did not include any T cell subset in this figure purposefully. In addition, Th2 cells do not express AHR (Veldhoen et.al., 2008, Nature). As explained also in response to comment 6 of Reviewer 5, we respectfully refrain to add other skin conditions to this figure as psoriasis and AD are the two inflammatory skin diseases where the AHR pathway has been more deeply investigated, hence we chose them as the primary focus of this review, as we now state in the abstract (page 1, line 19) and in the main text (page 7, lines 259-261).
“A summary of the current knowledge on the AHR pathway in the two inflammatory skin diseases where it has been studied most extensively, i.e. psoriasis and atopic dermatitis, versus healthy skin is displayed in Figure 3.”
Comment 6b:
Also, in figure 1 make sure to place the legend about Biorender at the end of legend (after abbreviations).
Response 6b:
We thank Reviewer 2 for bringing this to our attention. This has been changed for all figures (page 3, line 109; page 5, lines 200-201; page 12-13, line 470).
Reviewer 3 Report
Comments and Suggestions for Authors
Review on The Aryl Hydrocarbon Receptor (AHR): Peacekeeper of the 2 Skin by Hannah R Dawe and Paola Di Meglio.
This manuscript is an excellent, detailed review devoted to the patho-/physiological role of AHR in skin with the focus on Psoriasis and Atopic dermatitis.
Minor comments:
1) Affiliations should be in the same font size.
2) “Acid” in “Acidic pH in Stomach Acid” may be avoided in Figure 2
3) (a) or [a] in Figure 2 legend should be unified – for BaP vs DMBA
4) I recommend avoiding the use of a citation that refers to a conference abstract (here the reference number 88) where there is no traceable data for that abstract that would conclusively show that Tapinarof and TCDD do not compete for binding to AHR. Therefore, the whole sentence at lines 257-258 is pointless.
Author Response
Comment 1:
1) Affiliations should be in the same font size.
Response 1:
We thank Reviewer 3 for bringing this to our attention. This has been rectified (page 1, line 5).
Comment 2:
2) “Acid” in “Acidic pH in Stomach Acid” may be avoided in Figure 2.
Response 2:
We thank Reviewer 3 for bringing this to our attention. This has been changed in Figure 2 (page 5).
Comment 3:
3) (a) or [a] in Figure 2 legend should be unified – for BaP vs DMBA
Response 3:
We thank Reviewer 3 for bringing this to our attention. This has been changed as suggested (page 5, line 195).
Comment 4:
4) I recommend avoiding the use of a citation that refers to a conference abstract (here the reference number 88) where there is no traceable data for that abstract that would conclusively show that Tapinarof and TCDD do not compete for binding to AHR. Therefore, the whole sentence at lines 257-258 is pointless.
Response 4:
We respectfully believe that information from published abstracts to be important to acknowledge as it demonstrates what the field is moving towards in terms of new discoveries. We have added a statement that this data is from a conference abstract (page 7, line 281) and that additional work following this up would be needed (page 7, lines 283-284).
“Moreover, a published abstract from a conference has suggested that tapinarof and TCDD do not compete for binding to AHR, likely engaging via distinct binding pockets on the receptor [93, 94]; further studies should be undertaken to confirm this finding and how it relates to tapinarof’s mechanism of action.”
Reviewer 4 Report
Comments and Suggestions for Authors
The article is well organized but I have some questions:
(1) Give a brief comparison between Exogenous and Physiological ligands.
(2) In Figure 3, what are the expressions in Psoriasis and atopic dermatitis, respectively?
(3) References are in incorrect format.
Author Response
Comment 1:
(1) Give a brief comparison between Exogenous and Physiological ligands.
Response 1:
We thank Reviewer 4 for highlighting this important point. We have clarified the difference between exogenous and physiological ligands to provide more information on this important topic (page 4, lines 141-145).
“In contrast to exogenous ligands, which are either metabolized into or undergo bioactivation into carcinogenic metabolites [43], physiological ligands undergo biotransformation into non-toxic metabolites for excretion [23], a distinction with important clinical implications in terms of safety for their therapeutic application.”
Comment 2:
(2) In Figure 3, what are the expressions in Psoriasis and atopic dermatitis, respectively?
Response 2:
We thank Reviewer 4 for or bringing up this point which we hope to clarify here. As reported in the text (page 8, lines 325-343) and in Table 1 (page 9, line 356), there are conflicting reports about the expression of AHR and CYP1A1 being increased or decreased in psoriasis. This is displayed in Figure 3 as ↑↓? , with the figure legend explaining this notation (page 13, lines 460-461). AHR expression is consistently reported as increased in atopic dermatitis (Table 2, lines 429-430), which is displayed in Figure 3 as ↑, while CYP1A1 expression is again conflicting (Table 2, lines 430-434) and is displayed in Figure 3 as ↑↓?. We hope this clarifies this point.
Comment 3:
(3) References are in incorrect format.
Response 3:
We thank Reviewer 4 for bringing this to our attention. This has been rectified (pages 14-20, lines 517-790).
Reviewer 5 Report
Comments and Suggestions for Authors
The manuscript provides a comprehensive review of the aryl hydrocarbon receptor (AHR) and its critical role in maintaining skin homeostasis and modulating inflammation. The writing is clear, scientifically accurate, and well-referenced. It effectively balances detailed molecular mechanisms with clinical relevance, making it accessible to a broad scientific audience while maintaining the necessary depth for experts in the field.
It is known that the AHR signalling pathway plays an important role in various physiological processes, including the regulation of skin homeostasis, inflammation, and immune responses. It accomplishes this by regulating various aspects of skin function, including the immunological network, keratinocyte differentiation, skin barrier function, pigmentation, and responses to oxidative stress. Dysregulation of this pathway has been associated with various diseases, including cancer, immune disorders, and developmental abnormalities. The AHR signalling pathway plays a significant role in dermatoses, which are a group of skin disorders characterized by various symptoms, including inflammation, itching, redness, and abnormal cell proliferation.
Abstract
The abstract provides a concise yet thorough overview of the manuscript’s focus, emphasizing the physiological and therapeutic roles of AHR in skin health and disease. It appropriately highlights the relevance of tapinarof as a therapeutic agent, grounding the discussion in clinical applications. The abstract aligns well with the content of the manuscript.
Comment
- Consider explicitly stating the scope of the review in terms of the diseases covered, such as psoriasis and atopic dermatitis or HS, acne, to enhance clarity.
Introduction
The introduction effectively outlines the historical context and emerging importance of AHR, transitioning smoothly from its toxicological implications to its physiological roles. It provides a compelling rationale for the review.
Comment
- The section could briefly address gaps in knowledge or controversies in the field to better frame the subsequent discussion.
The AHR Pathway
This section is particularly well-done, offering a detailed molecular mechanisms explanation of AHR activation and regulation. The use of diagrams enhances the reader's understanding of complex pathways.
Comment
- The role of non-canonical pathways, such as interactions with other signaling pathways (e.g., EGFR), could be explored further to broaden the discussion.
AHR Ligands
The section provides a thorough overview of exogenous and endogenous AHR ligands, emphasizing their dual role in homeostasis and pathology. The inclusion of specific examples like coal tar and manuka honey is particularly engaging.
Comment
- Highlight potential clinical implications of differences between physiological and exogenous ligands, particularly regarding therapeutic targeting.
AHR in Healthy Skin
The section effectively details AHR’s role in maintaining skin barrier integrity and immune regulation. The discussion of keratinocyte differentiation and the influence of the microbiome is particularly compelling.
Comment
- The role of AHR in specific immune cells (e.g., Langerhans cells) is well-documented but could benefit from a clearer connection to therapeutic applications.
AHR in Inflammatory Skin Diseases
The discussion of psoriasis and atopic dermatitis is comprehensive, with a balanced presentation of preclinical and clinical data. The manuscript effectively integrates findings from molecular studies and clinical trials.
Comment
- The manuscript could address the potential role of AHR modulators in emerging inflammatory skin conditions, such as hidradenitis suppurativa or acne. The authors did not explain why they describe the role of AHR only in these two conditions. I think that expanding the manuscript with additional information on other inflammatory skin diseases would improve the quality of the work.
Conclusions and Perspectives
The conclusion effectively synthesizes the findings and emphasizes the therapeutic potential of AHR modulation. The call for further research into discrepancies in AHR activation is well-justified.
Comment
- Discuss the challenges of targeting AHR therapeutically, such as potential off-target effects or inter-individual variability in AHR expression and activation.
References
The references are extensive and relevant, reflecting the manuscript's thorough engagement with current literature.
Final Comments
This manuscript is an good contribution to the field, offering both depth and breadth in its exploration of AHR's role in skin health and disease. It is well-structured, scientifically rigorous, and highly relevant to researchers and clinicians.
Overall Recommendations:
- Minor refinements in framing and expanding certain sections would further enhance the manuscript.
- Emphasizing the translational aspects and addressing potential challenges of AHR-targeting therapies could provide additional value.
The manuscript is suitable for publication following minor revisions.
Author Response
Comment 1:
- Consider explicitly stating the scope of the review in terms of the diseases covered, such as psoriasis and atopic dermatitis or HS, acne, to enhance clarity.
Response 1:
We thank Reviewer 5 for this useful comment. We have revised the abstract as suggested to clarify that the main focus of the review on psoriasis and atopic dermatitis (page 1, line 19), and provide the rationale for this choice in the introduction (page 1, lines 37-38). Please see also response to comment 6 below.
“Here we review the current literature, describing AHR functions in skin health and disease, focusing on psoriasis and AD where the AHR pathway has been more extensively investigated…”
Comment 2:
- The section could briefly address gaps in knowledge or controversies in the field to better frame the subsequent discussion.
Response 2:
We concur with Reviewer 5 and have explicitly mentioned discrepancies in the literature regarding AHR expression in psoriasis and AD which warrant further investigation in the Introduction (page 1, line 40-41).
“…and identifying discrepancies in the literature regarding AHR expression in psoriasis and AD which warrant further investigation.”
Comment 3:
- The role of non-canonical pathways, such as interactions with other signaling pathways (e.g., EGFR), could be explored further to broaden the discussion.
Response 3:
We thank Reviewer 5 for this suggestion but have respectfully elected not to include this topic as non-canonical AHR signaling has been reviewed in detail recently by Sondermann et.al., 2023, Biochem Pharmacol, which we cite on page 2, lines 86-87, REF 26, not to duplicate the information.
Comment 4:
- Highlight potential clinical implications of differences between physiological and exogenous ligands, particularly regarding therapeutic targeting.
Response 4:
We thank Reviewer 5 for bringing up this important point. We have highlighted the difference between exogenous and physiological ligands in terms of their metabolism (bioactivation vs biotransformation) which has important clinical implications in terms of safety for their therapeutic application (lines 141-145).
“In contrast to exogenous ligands, which are either metabolized into or undergo bioactivation into carcinogenic metabolites [43], physiological ligands undergo biotransformation into non-toxic metabolites for excretion [23], a distinction with important clinical implications in terms of safety for their therapeutic application.”
Comment 5:
- The role of AHR in specific immune cells (e.g., Langerhans cells) is well-documented but could benefit from a clearer connection to therapeutic applications.
Response 5:
We thank Reviewer 5 for this suggestion. To the best of our knowledge there are no published studies investigating the connection between the role of AHR in LCs and therapeutic modulation of AHR. We have added a sentence to reflect this research gap in the field (line 251-252).
“How AHR’s role in LCs relates to the therapeutic activation of the pathway in humans remains to be elucidated.”
Comment 6:
- The manuscript could address the potential role of AHR modulators in emerging inflammatory skin conditions, such as hidradenitis suppurativa or acne. The authors did not explain why they describe the role of AHR only in these two conditions. I think that expanding the manuscript with additional information on other inflammatory skin diseases would improve the quality of the work.
Response 6:
As already stated in response to comment 6a of Reviewer 2, we focussed on psoriasis and atopic dermatitis as exemplar inflammatory skin diseases, where the AHR pathway has most deeply investigated. To address these comments, we now clarify the focus on psoriasis and atopic dermatitis in the abstract (page 1, line 19), in the introduction (page 1, lines 37-38) and in the main text (page 7, lines 259-261).
“…and discuss how the pathway is dysregulated in psoriasis and atopic dermatitis…”
“Here we review the current literature, describing AHR functions in skin health and disease, focusing on psoriasis and AD where the AHR pathway has been more extensively investigated…”
“A summary of the current knowledge on the AHR pathway in the two inflammatory skin diseases where it has been studied most extensively, i.e. psoriasis and atopic dermatitis, versus healthy skin is displayed in Figure 3.”
Moreover, we already mention other skin conditions where there are published evidence of the pathway to be involved, namely HS, where we include only study investigating the AHR pathway in HS showing dysregulation in tryptophan metabolism and impaired AHR activation (Guenin-Mace et.al., 2020, JCI Insight, Ref 134; Page 13, lines 480-482). Moreover, we mention case reports where therapeutic AHR activation by tapinarof has been used in vitiligo (Ref 135), seborrheic dermatitis (Refs 136, 137), and palmoplantar keratoderma (Ref 138) (page 13, lines 482-484).
“For example, dysregulation in tryptophan metabolism and a reduction in AHR activation, shown as reduced expression of CYP1A1, AHRR, and CYP1A2, has been detected in the skin of patients with hidradenitis suppurativa (HS) [134]. Moreover, case reports of efficacious off-label use of tapinarof in vitiligo [135], seborrheic dermatitis [136, 137], and palmoplantar keratoderma [138]…”
In response to this Reviewer and Reviewer 2 comments 3 and 4, we have now also included information on contact dermatitis, acne, as well as frontal fibrosing alopecia (FFA), as additional examples of skin diseases where evidence in the literature support a role for the AHR pathway and its therapeutic modulation (page 13, lines 486-503).
“Other inflammatory skin diseases like contact dermatitis, acne, and frontal fibrosing alopecia (FFA) may also benefit from AHR modulation as a possible therapeutic strategy. Hapten-induced contact hypersensitivity (CHS) is an established mouse model of allergic contact dermatitis where the skin is sensitised to haptens like DNFB or FITC applied to the dorsal skin, followed by hapten exposure on the ear five days later [139]. A potential role for AHR in contact dermatitis has been explored using these models, with AHR-/- mice experiencing a reduced CHS response compared to wild-type [85] using FITC-induced CHS. Conversely, using a DNFB-induced CHS model, indirect AHR activation using 4-n-nonylphenol (NP) also reduced the CHS response [140]. The only human study of contact dermatitis conducted in 1982 reported reduced activity of aryl hydrocarbon hydroxylase (the historical term for CYP1A1) in both irritant and allergic contact dermatitis patients [141]. Indoles have been shown to attenuate inflammation in an AHR-dependent manner in a rat model of acne vulgaris [142], a chronic inflammatory disease of the pilosebaceous unit of the skin [143], not to be confused with chloracne, which is the non-inflammatory skin condition resulting from dioxin exposure [32]. Finally, the identification of a putative causal missense variant in CYP1B1 for FFA [144] provides genetic evidence for an involvement of the AHR pathway, suggesting that AHR modulation may be a useful therapeutic strategy for FFA.”
Comment 7:
- Discuss the challenges of targeting AHR therapeutically, such as potential off-target effects or inter-individual variability in AHR expression and activation.
Response 7:
We now discuss the potential link between inter-individual variability in AHR expression and clinical response to tapinarof. We have added this to the psoriasis paragraph rather than to the conclusions as it fit more with our discussion points (page 8, lines 343-346).
“Importantly, differential expression of AHR between studies may suggest inter-individual variability in response to tapinarof because of higher or lower AHR expression in these patients. Real-world data on tapinarof’s use in the clinic will be crucial to understand whether this is the case.”